# Effect of the Mediterranean Diet (MeDi) on the Progression of Retinal Disease: A Narrative Review

**DOI:** 10.3390/nu16183169

**Published:** 2024-09-19

**Authors:** Oualid Sbai, Filippo Torrisi, Federico Pio Fabrizio, Graziella Rabbeni, Lorena Perrone

**Affiliations:** 1Laboratory of Transmission, Control and Immunobiology of Infections (LTCII), LR11IPT02, Institut Pasteur de Tunis, Tunis 1068, Tunisia; oualid.sbai@pasteur.tn; 2Faculty of Medicine and Surgery, University KORE of Enna, 94100 Enna, Italy; filippo.torrisi@unikore.it (F.T.); federicopio.fabrizio@unikore.it (F.P.F.); graziella.rabbeni@unikore.it (G.R.)

**Keywords:** Mediterranean diet, retinopathy, diabetic retinopathy, age-related macular degeneration, glaucoma, Nrf2

## Abstract

Worldwide, the number of individuals suffering from visual impairment, as well as those affected by blindness, is about 600 million and it will further increase in the coming decades. These diseases also seriously affect the quality of life in working-age individuals. Beyond the characterization of metabolic, genetic, and environmental factors related to ocular pathologies, it is important to verify how lifestyle may participate in the induction of the molecular pathways underlying these diseases. On the other hand, scientific studies are also contributing to investigations as to whether lifestyle could intervene in modulating pathophysiological cellular responses, including the production of metabolites and neurohormonal factors, through the intake of natural compounds capable of interfering with molecular mechanisms that lead to ocular diseases. Nutraceuticals are promising in ameliorating pathophysiological complications of ocular disease such as inflammation and neurodegeneration. Moreover, it is important to characterize the nutritional patterns and/or natural compounds that may be beneficial against certain ocular diseases. The adherence to the Mediterranean diet (MeDi) is proposed as a promising intervention for the prevention and amelioration of several eye diseases. Several characteristic compounds and micronutrients of MeDi, including vitamins, carotenoids, flavonoids, and omega-3 fatty acids, are proposed as adjuvants against several ocular diseases. In this review, we focus on studies that analyze the effects of MeDi in ameliorating diabetic retinopathy, macular degeneration, and glaucoma. The analysis of knowledge in this field is requested in order to provide direction on recommendations for nutritional interventions aimed to prevent and ameliorate ocular diseases.

## 1. Introduction

Eye diseases have a serious influence on overall quality of life, health, and the possibility of obtaining an appropriate education, developing one’s own capabilities in the working environment, and contributing to sustainable development, and these conditions ultimately affect the economy and well-being across society. Indeed, the effects of visual impairment are not limited to daily routines, while they have a negative effect on both psychological and cognitive development by affecting educational and employment possibilities [1]. Moreover, impaired vision is a predisposing factor for various health disorders, including cardiovascular illness, dementia, cancer, and depression [2].

It has been estimated that about 600 million people have visual impairment and the large majority of these patients can be cured only with high-cost therapies [3]. Notably, 43 million people exhibit partial or complete blindness, while about 500 million patients show poor socio-economic conditions that block them from ameliorating their vision impairment because they cannot afford the cost of reading glasses [3]. Indeed, vision impairment shows a more elevated prevalence in women and people living in rural areas, creating a vicious circle that exacerbates poverty and the low educational level of such people, affecting their ability to improve their socio-economic conditions. 

Poor educational and socio-economic conditions are associated with a poor food supply. Interestingly, inadequate dietary patterns are associated with an enhanced risk of developing eye diseases. Considering the high cost needed to cure vision impairment, a good diet can exert preventive action against eye diseases that can lead to blindness. However, in highly developed countries, a good standard of living does not necessarily ensure an amelioration of the quality of life, especially in the quality of aging. Indeed, despite higher levels of education, poor lifestyle choices can promote inadequate nutrition, increasing the risk of chronic and degenerative diseases [4]. The Mediterranean diet (MeDi) is considered a healthy diet that is very efficient for the prevention of several diseases [5]. It is characterized by several healthy components and is not expensive; thus, it is a sustainable resource for the prevention of visual impairment. Herein, we will summarize the effect of MeDi in preventing retinal disorders. We will also focus on the Nrf2 pathway in mediating the advantageous effects promoted via MeDi. To summarize the data contained in this narrative review, we used the following PubMed analysis method to collect published data. (i) We typed in PubMed “Mediterranean diet, diabetic retinopathy”, providing 25 articles. We analyzed these articles and the articles in their citations, resulting in the analysis of data derived from 882 articles. (ii) We typed in PubMed “Mediterranean diet, macular degeneration”, providing 36 articles. We analyzed these articles and those included in their citations, resulting in the analysis of data derived from 688 articles. (iii) We typed in PubMed “Mediterranean diet, glaucoma”, providing 10 articles. We analyzed these articles and those included in their citations, resulting in the analysis of data derived from 623 articles. (iv) We typed in PubMed “Mediterranean diet Nrf2”, resulting in 31 publications that we analyzed. (v) We typed in PubMed “retinopathy, Nrf2”, resulting in 402 publications that we analyzed.

## 2. Visual System and Conditions Leading to Retinal Diseases

The visual system consists of the coordinated interaction of visual pathways between the eyes and the brain, and it also involves the tissues associated with the eyes. The cornea and the lens of the eyes direct the light onto the retinal photoreceptors that, in turn, transform the light-induced stimulation into neuronal impulses. Finally, these impulses are translated into tri-dimensional images in the brain. 

The architecture of the retina is highly ordered and conserved among all vertebrates. It is constituted by five types of neurons dispersed in three nuclear layers that are separated by two plexiform layers formed by synaptic interactions. Photoreceptors are localized in the outer nuclear layer (ONL); different types of interneurons (bipolar, amacrine, and horizontal cells) are present in the inner nuclear layer (INL); the ganglion cell layer (GCL) contains the retinal ganglion cells (RGCs) and dislocated amacrine cells. Photoreceptors are stimulated by the light, producing electrochemical signals that are transmitted by synapses with bipolar and horizontal cells in the outer plexiform layer (OPL). The inner plexiform layer (IPL) contains the synapses from ganglion cells to amacrine and bipolar cells. The projections of ganglion cell axons constitute the optic nerve, which transmits the signals from the eye to the brain for visual processing. Vision derives from specific patterns of connections from each type of neuron, which leads to the formation of different ganglion cells that show specific sensitivity to different stimulations, including stationary or moving objects, color contrast, and edges. In the retina are present various types of glial cells. The predominant type is the Muller glia that interacts with all neuronal types. The Muller glia modulates neuronal and microglia function through various secreted molecules, such as neurotransmitters [6], and this glial type seems to be involved in retina regeneration approaches [7]. Finally, astrocytes are mostly present in the IPL, together with the microglia, and they modulate retina homeostasis and are involved in promoting inflammation in retinal diseases [8].

Vision can be affected by genetic mutations, age, malnutrition, environment, and lifestyle [9,10,11]. Eye diseases affecting vision and leading to retinopathy are increasing. Retinal diseases include a range of genetic and non-genetic disorders. Genetic retinal diseases show an incidence of 1 to 3000 individuals [12], and more that 340 genes are implicated in those disorders [13]. Non-genetic retinal diseases can be modulated by genetic factors, but are mostly influenced by environmental factors, lifestyle, infections, and aging. Age-related macular degeneration is one of the most prevalent non-genetic retinal disorders and leads to central vision loss caused by the accumulation of deposits and by retinal pigment damage in the non-vascular variant, or by increased vascular growth in the neovascular variant [14]. Diabetic retinopathy (DR) is also a common retinal disease characterized by alterations of the retinal blood vessels [15], leading to inflammation, gliosis, and neuronal injury in the GCL [16,17]. Glaucoma is characterized by optic nerve structural damage with axonal loss, and RGC apoptosis [18]. Herein, we will focus on the following retinopathies: diabetic retinopathy (DR) [19], age-related macular degeneration [20], and glaucoma [3], as well as the effect of MeDi in preventing/ameliorating these ocular disorders.

## 3. The Mediterranean Diet (MeDi)

The Mediterranean Diet (MeDi) is not limited to a dietary pattern, while it includes a specific lifestyle. The pattern of lifestyle and diet at the basis of the MeDi originated a long time ago and it is the combination of various cultures that characterized the Mediterranean region: Roman, Greek, Phoenician, Arabic, and other cultures that shared their cultural and nutritional patterns, influencing their lifestyle [21]. In 2010, UNESCO acknowledged MeDi as an Intangible Cultural Heritage of Humanity and developed the model of the food pyramid (Figure 1) in order to communicate the MeDi model to people and health professionals [22]. 

MeDi is constituted not only by a dietary pattern, but also sustainable food production, conviviality, and an active lifestyle, including daily social activity that promotes moderate physical activity as well as an appropriate time for rest [23]. The MeDi dietary pattern includes the daily utilization of fresh vegetables and fruits, nuts and seeds, whole grains, eating legumes several times/week, the utilization of extra virgin oil as source of cooking and seasoning fat, herbs, and spices for flavoring, resulting in low salt consumption, low intake of cakes and desserts, two to three servings/week of fish and seafood, two to four servings of eggs/week, daily consumption of low-fat dairy products (mostly yogurt), the consumption of red meat no more than once a week, drinking water instead of other beverages, and moderate consumption of wine, mostly red wine, during meals [22]. However, there are some differences between MeDi components in different countries of the Mediterranean area that are linked to cultural, economic, and religious differences between the countries [24], as described below. For example, there are differences in alcohol consumption. The Greek MeDi includes general alcohol consumption, while French and Italian MeDi includes the consumption of red wine. In particular, the Lebanese MeDi shows several differences compared to the other Mediterranean countries; it includes dried fruits and burghul, which are traditional food in Lebanon, while red meat, fish, and alcohol are not included in the Lebanese MeDi [25].

## 4. MeDi Scoring

It has been shown that MeDi decreased the risk of several chronic illness [5]. MeDi scoring has been used in clinical observational investigations aimed at analyzing the impact of the adherence to MeDi on the progression of various diseases. Several scoring systems have been used with the aim of defining MeDi adherence (Figure 2). For the qualitative interpretation of the data, the adherence to MeDi is classified into classes: low, moderate, and high adherence. The first study analyzing the impact of MeDi on the survival of the Greek population was published in 2003 by Trichoupoulo and colleagues [26]. The authors analyzed the adherence to MeDi in a population-based prospective investigation using a food frequency questionnaire and the dietary habits have been classified in a 10-point Mediterranean diet scale (MDS), with a score ranging from 0 to 9, where 9 represents the highest adherence. The scale included the consumption frequency of the most salient foods typical of the Greek diet. In 2006, Panagiotakos et al. defined MedDietScore [27], a five-point scale based on the frequency of consumption of 11 principal constituents of MeDi (whole cereals, vegetables, fruits, legumes, potatoes, fish, poultry, red meat, olive oil, full-fat dairy products, and alcohol). A score of 5 was considered the most adherent to MeDi (low alcohol and red meat consumption/day). Buckland and colleagues created rMed score in 2010 [28], which is an 18-point linear scale containing nine components of the diet. A score of 18 was considered the most adherent to the MeDi. Moreover, the rMed score was divided into low (0–6), medium (7–10), and high (11–18), called tertiles. Schroder et al. defined the Mediterranean Diet Adherence Screener (MEDAS) scoring [29]. MEDAS is composed of twelve questions related to food consumption frequency, and two questions related to food consumption habits. Each question is scored 0 or 1. In particular, one point is assigned for utilizing olive oil as the main kind of fat for cooking, consuming white meat instead of red meat, or for eating (1) 4 or more tablespoons (1 tablespoon = 13.5 g) of olive oil/d (utilized for frying, seasoning, meals consumed far from home, etc.); (2) two or more servings of vegetables/d; (3) three or more servings of fruit/d; (4) <1 portion of red meat or sausages/d; (5) <1 portion of animal fat/d; (6) <1 cup (1 cup = 100 mL) of sweet beverages/d; (7) seven or more portions of red wine/week; (8) three or more portions of pulses/week; (9) three or more portions of fish/week; (10) less than two commercial desserts/week; (11) three or more portions of nuts/week; or (12) two or more portions/week of a traditional plate composed of tomato sauce, garlic, onion, or leeks sautéed in olive oil. When these conditions were not met, 0 points were assigned for the category. Thus, MEDAS scores range from 0 to 14, with 14 corresponding to the highest adherence to MeDi [29]. In 2013, Agnoli et al. described the Italian Mediterranean Index (IMI) [30]. IMI was built on eleven food categories, including six typical Mediterranean food categories (pasta, fish, legumes, Mediterranean vegetables, fruits, and olive oil), four non-Mediterranean food items (sweet beverages, butter potatoes, and red meat), and alcohol. Subjects included in the third tertile of consumption of each characteristic Mediterranean food category were assigned a score of 1, while the others were assigned a score of 0. For non-Mediterranean food items, a score of 1 was assigned for subjects included in the first tertile of consumption and 0 for the others. Concerning alcohol consumption, 1 point was given for the subjects drinking up to 12 g per day, and 0 for abstainers or subjects drinking more than 12 g of alcohol/day. Scores varied between 0 and 11. In 2015, Naja et al. described the Lebanese Mediterranean Index (LMD) [25]. LMD score analyzed the eating frequency of the following categories of foods: vegetables, fruits, dried fruits, legumes, burghul, olive oil, starchy vegetables, eggs, and dairy products. The score was evaluated using a 61-item semi-quantitative questionnaire of food frequency and then compared to the scoring results derived from other studies. The Mediterranean diet scale (MDS) was realized in order to analyze the adherence to the Mediterranean diet in nine European countries (Denmark, Germany, Greece, France, Italy, Spain, the Netherlands, UK, and Sweden) and analyzed nine food categories. Subjects received a score of 1 when their consumption of legumes, vegetables, cereals, fish, and fruits was lower compared to the sex-specific average intake, while they received a score 0 in all the other cases. The opposite scoring was given for the categories of meat and dairy. Concerning alcohol consumption, men drinking 10–50 g/day and women drinking 5–25 g/day received a score of 1, while all the other cases received a score of 0. The ratio of monounsaturated fat (MUFA) and polyunsaturated fat (PUFA) to saturated fat (SFA) was also considered. This scoring ranged between 0 and 9 [25]. Monteagudo and colleagues defined the Mediterranean Diet Scoring System (MDSS) in 2015 [31]. MDSS was calculated from the data obtained using a questionnaire with 129 items divided into 11 food categories (cereals, fruit, vegetables, fish, eggs, meat, fats, commercial foods, sauces, alcohol-free drinks, and alcohol). MDSS was created according to the Mediterranean Diet Pyramid and the recommended eating/drinking frequency of the various categories, and the range of the scores is 0–24 for adults and 0–23 for adolescents (eliminating alcoholic beverages). The points were assigned considering the consumption of food categories in accordance with the recommended servings: a score of 3, 2, or 1 for recommendations calculated in times/meal, times/day, or times/week, respectively. Thus, this scoring provided higher relevance to foods recommended to be consumed at every meal (fruit, vegetables, olive oil, cereals), followed by foods recommended to be consumed daily (dairy products, dried fruit, and nuts), and lastly, foods recommended to be consumed once a week (potatoes, eggs, legumes, white meat, fish, red meat, desserts). In adults, 1 point was assigned for alcohol consumption corresponding to one (women) and two (men) glasses of wine or beer. Sofi and colleagues developed the MeDi-Lite score in 2017 that analyzed nine food groups and compared the results with the MDS score, demonstrating that the higher range score provided an increased sensibility and specificity compared to MDS scoring [32].

These differences in the scoring result in an increased difficulty in comparing the data obtained from different observational, retrospective, and prospective clinical studies (Figure 2). Obeid and colleagues compared the different scoring systems by grouping them in the tertiles that defined the low, medium, and high adherence to the MeDi [33]. Moreover, certain studies provide the scores as consumption of food groups in grams/day, while others used the servings/week. This difference results in an additional complication when analyzing the amount of defined nutrients in different studies, according to published conversion data providing the amount of certain nutrients in defined food. The conversion from servings/week to grams/day has been published in order to solve this complication and extrapolate the content of dietary/nutritional compounds from different clinical studies [34].

## 5. Natural Molecules Enriched in the MeDi

MeDi is enriched in various nutraceuticals that produce beneficial outcomes for health.

### 5.1. Phenolic Compounds

Phenolic compounds are plant-derived micronutrients produced and secreted by plants following infection by pathogens or ultraviolet radiation. Polyphenols are classified according to the phenol rings they contain and the association between these rings and carbohydrates or organic acids. Some red fruits, black radish, tea, and onion contain tannins and simple phenols, such as gallic acid, which derives from benzoic acid. Derivatives of cinnamic acid are more common, and frequently, they are glycosylated. This group includes flavonoids, stilbene, and lignans [35]. Flavonoids have been extensively studied and they include flavonols (e.g., quercetin), flavones, flavonones, isoflavonones (e.g., gynestein), and anthocyanins. Resveratrol (RV) is the most studied stilbene and its antioxidant function is well characterized as well as its role in glucose homeostasis [36]. Polyphenols are mostly derived from fruits, vegetables, tea, and red wine (RV). They show an antioxidant activity that exerts a beneficial effect against several chronic diseases. Moreover, they are modified by the gut microbiota, opening the way for the study of their role on the metabolism of the microbiota and the subsequent effect on the human body [37,38]. Blackberries, blueberries, strawberry, kiwi, cherry, apricot, apple, pear, and all other types of fruits, including nuts, contain high levels of polyphenols. Whole grains also contain phenolic compounds, which are lost in refined grains.

### 5.2. Isoprenoids

This group includes carotenoids, saponins, tocotrienols, tocopherols, and simple terpenes, and they are contained in vegetables and fruits [39]. Carotenoids, in particular β-carotene, lycopene, zeaxanthin, and lutein, exert pro-vitamin A activity and are potent antioxidants [39]. Indeed, they are beneficial against cancer and neurodegenerative diseases, preventing cataracts. They also prevent age-related diseases by enhancing immune system activity [40]. Lycopene modulates the redox signaling that regulates gene expression [41]. Tocopherols and tocotrienols are two isoforms of vitamin E; they are found in plants and seeds. They are potent antioxidants, preventing DNA damage. They also exert anti-inflammatory action and are known to be protective against cancer, cardiovascular diseases, and neurodegeneration [42].

Saponins are also derived from plants and the most known saponins are the ginsenosides, derived from the ginger root. They have anti-cancer activity [43]. Green vegetables show high levels of lutein, β-carotene, and β-cryptoxanthin; carrots and pumpkins show high content of α-carotene; oranges, red bell peppers, broccoli, green vegetables, and potatoes together with carrots contain β-carotene. β-cryptoxanthin is found in tropical fruits like papaya. Tomato, watermelon, and pink grapefruit contain lycopene. Green vegetables including spinach, Brussels sprouts, broccoli, and peas are a source of lutein, while egg yolks and corn contain high levels of zeaxanthin. 

### 5.3. Carbohydrates

Carbohydrates are grouped according to their digestibility in the gastrointestinal tract. Starch and fructans are hydrolyzed and adsorbed in the small intestine. On the contrary, β-glucans cellulose, hemicellulose pectin, and lignin cannot be digested in the small intestine and they are transformed in the large intestine through bacterial fermentation. Several investigations show that β-glucans modulate cholesterol metabolism, are beneficial against colorectal cancer, reduce constipation, and promote the growth of the gut microbiota [44]. Moreover, β-glucans lower the blood level of low-density lipoprotein (LDL) cholesterol particles by enhancing the fecal excretion of bile acids, resulting in an increased transformation of cholesterol in bile acids in the liver [45]. β-glucans ameliorate glycemic rate and the insulin response [46] by promoting insulin signaling in the liver [47]. Pectin is beneficial in promoting lipid and cholesterol metabolism [48], and diminishes intestinal infections by reducing the growth of pathogenic bacteria [49]. Starches that cannot be digested in the small intestine are defined as fibers and are enriched in whole grains. They decrease postprandial glucose and insulin levels [50], and lower cholesterol and triglyceride concentrations [51]. 

### 5.4. Proteins

Eggs, fish, meat, and dairy products provide high-quality proteins; beans and peas contain good-quality proteins, and grains are a moderate source of proteins. MeDi is defined according to a low consumption of meat, a moderate intake of fish, and a high consumption of beans and grains. The low intake of meat has been proposed as beneficial against various chronic diseases, including cardiovascular disease, diabetes, and cognitive dysfunction, in particular Alzheimer’s disease. Several studies demonstrate that meat contains high levels of Advanced Glycation Endproducts (AGEs), which induce both oxidative stress and inflammatory response [52]. Dietary AGE intake correlates with enhanced incidence of chronic disorders, e.g., Alzheimer’s disease [34]; thus, MeDi is also beneficial in preventing chronic illness by providing low AGE content [34]. AGEs participate in the progression of chronic diseases by inducing oxidative stress and by activating the Receptor For Advanced Glycation Endproducts (RAGE), which exerts a major role in ocular diseases characterized by retinopathy, including diabetic retinopathy, age-related macular degeneration, and glaucoma [16,53] (Figure 3).

### 5.5. Lipids and Fatty Acids

Linoleic acid (LA) is an essential poly unsaturated fatty acid (PUFA), important for the formation of phospholipids that constitute the plasma membranes and for the formation of the lipoprotein particles that regulate cholesterol homeostasis. LA is present in several components of the MeDi, including sunflower oil, grape seed oil, safflower oil, walnuts, salmon, chia seeds, and sardines. LA derivatives constitute the omega-6 fatty acids. In particular, they participate in the formation of high-density lipoprotein (HDL). High dietary LA intake decreases the risk of cardiovascular diseases [54]. Dietary alpha-linoleic acid (LNA) intake is also protective against cardiovascular diseases [55,56]. The conversion products of LNA are present in fish oil. They are eicosapentaenoic acid (EPA) and docosahexaenoic acid (DHA) and are the omega-3 fatty acids. Their dietary intake and fish consumption decrease the risk of cardiovascular diseases [56]. Both LA and LNA undergo several steps of desaturation (Δ6 and Δ5 desaturases) and elongation, generating a great number of metabolites, including arachidonic acid (ARA), from LA, and eicosapentaenoic (EPA) and docosahexaenoic (DHA) acids, from LNA. ARA generates the commonly known eicosanoids (ECs) anandamide (AEA) and 2-AG, while eicosapentaenoyl ethanolamine (EPEA) and docosahexaenoyl ethanolamine (DHEA) are produced from EPA and DHA, respectively, recently recognized as weaker ECs. DHEA is also known as synaptamide, a trophic factor that improves cognitive parameters in the nervous system. It is a metabolite generated on the omega-3 arm [57].

Several investigations demonstrate that the ratio between omega-3 and omega-6 fatty acids is relevant for preventing cardiovascular diseases. Indeed, eicosanoids derived from omega-6 PUFA promote inflammation, while eicosanoids derived from omega-3 PUFA have an anti-inflammatory function. Omega-3 fatty acid intake due to the consumption of fatty fish or fish oil exerts a healthy effect by decreasing the risk of cardiovascular diseases, rheumatoid arthritis, cancer, inflammatory bowel disease, and psychiatric and neurodegenerative diseases [58]. Omega-6 PUFA is contained in extra virgin olive oil. PUFAs are contained also in nuts, at high levels in pistachios, and in several vegetable oils derived from safflower, grape, sunflower, wheat germ, pumpkin seeds, sesame, and others [59]. 

### 5.6. Vitamins

We already described the function of vitamin E as an anti-inflammatory and antioxidant compound.

Vitamins A and D play an essential function in modulating the immune system [60]. The precursors of vitamin A (carotenoids) are pigments contained in vegetables and fruits, and they stimulate the immune system by promoting cell signaling pathways. The MeDi provides high levels of carotenoids with fruit and vegetables (e.g., tomatoes, leafy green vegetables, melons, carrots, bell peppers). Vitamin C modulates other antioxidant systems, including vitamin E, acting as an antioxidant, and can be found in fruits and vegetables.

Shellfish consumption provides high levels of vitamin B12. Rice, seaweed, soybeans, sesame seeds, peanuts, brown rice, and rye bread are great sources of vitamin B1. Spinach, avocados, and apricots provide high levels of vitamin B6 [61]. The vitamin B family is present in milk, cheese, eggs, fish, leafy vegetables, and chicken and exerts both antioxidant and neuroprotective actions [62].

### 5.7. Melatonin

High content of melatonin is present in fish, milk, eggs, seeds, and pistachios, providing neuroprotection and counteracting elevated intraocular pressure (IOP) [63].

### 5.8. Saffron

Saffron is also known as *Curcuma longa* and shows anti-inflammatory and antioxidant activity [64]. It contains curcumin and promotes several health benefits, such as helping to control diabetes, promoting weight loss, and preventing cardiovascular diseases. It inhibits pro-inflammatory cytokine release and modulates the composition of the gut microbiota [65].

### 5.9. Taurine

Taurine is the most copious amino acid in the retina of mammals [66] and is implicated in retinal survival [67]. Dietary taurine can be found in seafood, turkey, and seaweed [66].

### 5.10. Palmitoyethanolamide (PEA)

PEA is an N-acetylethanolamine cell-protective lipid present in various foods and in several living organisms. High concentrations of PEA are present in egg yolk. PEA shows anti-inflammatory and retina-protectant activity [68].

**Figure 3 nutrients-16-03169-f003:**
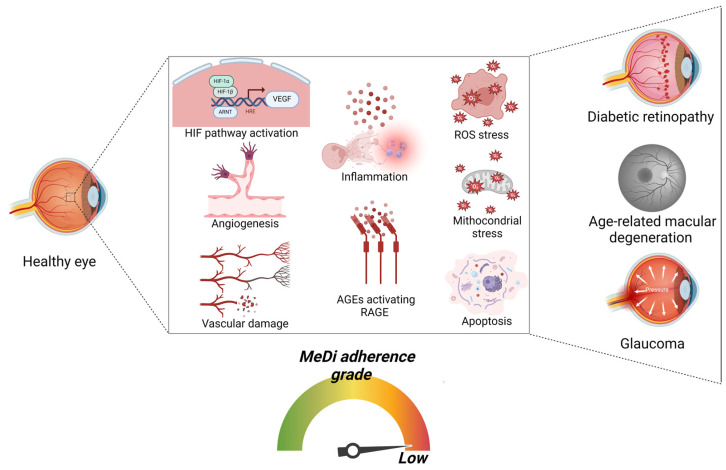
**Pathways and cellular alterations implicated in diseases characterized by retinopathy (DR, AMD, and glaucoma) and correlation with low MeDi adherence.** Scheme representing the progressive damage and the signaling pathways implicated in retinopathy and associated with low MeDi adherence. Initially, the activation of HIF pathway is observed, leading to VEGF expression, resulting in angiogenesis and subsequent vascular damage. Then, the formation of AGEs stimulates RAGE activation and subsequent inflammation. Ultimately, chronic excessive ROS formation and oxidative stress synergically lead to mitochondrial stress and apoptosis. These pathways are implicated in promoting the risk of DR, AMG, and glaucoma (Created with BioRender.com, Licensing and Agreement number XA2769UB0O).

## 6. MeDi and Stress Response Involved in Retinal Diseases: Focus on Nrf-2 Pathway

Retinal diseases share similar cellular and molecular pathways, involving inflammation, immune response, and neurodegeneration [69]. Notably, the characteristic of retinal disorders is an alteration of the balance between the formation of reactive oxygen and nitrogen species (ROS and RNS, respectively) and the induction of the antioxidant systems, leading to oxidative stress [70]. Indeed, the eye is subjected to both endogenous and environmental damaging factors, resulting in an increased sensitivity to ROS and RNS. Furthermore, age-related diseases characterized by an excess of ROS and RNS production, such as diabetes, are risk factors for retinal diseases [71,72]. Notably, the Nrf-2 pathway seems to be involved in modulating the effect of diet in the prevention of retinopathy (Figure 4). Metabolic disorders can lead to stressful conditions to which the organism must respond to reestablish homeostasis. It is important to identify molecular pathways that signal metabolic changes [73]. The stress response is a set of complex cellular and molecular signals that can be modulated by endogenous and exogenous factors. The correlation between stress and the promotion of chronic degenerative pathologies affecting various organs and systems, including the cardiovascular and nervous systems, has been widely discussed and continues to be a topic of research [74]. Exogenous factors, such as nutrients, can influence this process, while endogenous factors can be grouped into metabolites, hormones, and several molecules that include reactive oxygen species (ROS) and/or reactive nitrogen species (RNS) [75]. Food intake and energetic metabolism can modify the production of endogenous factors by altering the response to stress [76]. The primary molecular pathway that governs the stress response is the nuclear factor erythroid-2-related factor 2 (Nrf2) pathway that plays a crucial role in regulating cellular homeostasis by managing oxidative stress and detoxification [77]. Nrf2 mitigates oxidative damage via the transcriptional activation of antioxidant response elements (AREs), thereby promoting the antioxidant response process. A key domain of Nrf2 is Neh2, which interacts with Kelch-like ECH-associated protein 1 (Keap1). Under normal conditions, this interaction inhibits the transcriptional activity of Nrf2 by ubiquitination and degradation through the proteasome [78]. Disrupted homeostasis promotes excessive production of ROS and RNS that can activate immune cells to release proinflammatory factors. This activation triggers the transcriptional activation of ARE, regulating downstream antioxidant enzymes and various neuroprotective genes that impede oxidative stress and neuroinflammation, thereby blocking the onset of neurological disorders and the subsequent pathological processes [79].

In the framework of metabolism and nutrition, dietary energy restriction and adhering to the MeDi with the consumption of bioactive nutrients are the most studied approaches for regulating Nrf2 activity [80] (Figure 4). Dietary energy restriction, through either chronic or intermittent calorie reduction, increases Nrf2 activity. This creates an energetic stress in neurons that activates the Nrf2 pathway, leading to numerous health benefits, and increases longevity, promoting the prevention of neurological disorders [81]. Energetic dysfunctions and metabolite production can be correlated in the generation of a state of stress that involves different structures of the nervous system, including the eye [82]. Indeed, when the metabolite content is not sufficient for the energy demand, it increases the risk of retinal neuron death. Mitochondria are the center of energy supplementation and play an essential role in ATP generation and in sustaining redox homeostasis, and they drive the fate as waste of several types of metabolites [83]. Mitochondrial alteration is involved in the pathophysiology of several neurodegenerative diseases of the retina because the retina is highly susceptible to oxidative stress [84].

Nrf2 exerts a major function in promoting mitochondrial quality control and regulating fundamental aspects of mitochondrial function, such as energy production, ROS management, calcium signaling, and the induction of cell death [85]. Additionally, Nrf2 plays an essential function in modulating retinal oxidative stress. The retina shows a high metabolic activity and elevated oxygen consumption. For this reason, the retina is highly subjected to enhanced ROS production [86]. Studies indicate that there is reduced activity of the mitochondrial electron transport chain (ETC) in the aged retina, resulting in ROS augmentation and retinal damage. Impaired mitochondrial function leads to cell death and retinal degeneration. Moreover, the aged retina shows a reduced number of mitochondria, as well as altered mitochondrial activity and morphology compared to the healthy retina. In order to counteract the oxidative stress-induced damage, the retina relies on a crucial antioxidative defense system. Nrf2 is central to modulating the antioxidative stress response, especially against stressors including aging, inflammation, and sunlight exposure [86]. The induction of the Nrf2/Keap1/ARE cascade is considered a key target for neuroprotection in retinal ganglion cells [87]. Sox2 overlapping transcript (Sox2OT), a long non-coding RNA, highly expressed in the human brain, is involved in retinal ganglion cell apoptosis mediated by high glucose-induced reduction. It also induces Nrf2 nuclear translocation, determining HO-1 protein expression [87]. Dysregulation of the Keap1-Nrf2 cascade is also implicated in diabetic retinopathy. Activation of the Nrf2, MAPK, and NFκB signaling pathways effectively alleviates the ocular symptoms of diabetic retinopathy caused by ROS [88]. Moreover, Nrf2 expression in neurons also aids in detoxifying accumulated ROS by blocking mitochondrial complex II, suggesting that Nrf2 can protect neurons from the damage induced by dysfunctional mitochondria [89]. Moreover, Nrf2 plays a vital role in enhancing the antioxidant response, preserving the retina from ROS-induced cell damage. 

Several investigations underline the important role of natural products in counteracting oxidative stress by modulating Nrf2 function. Bioactive natural molecules present in food are essential in regulating Nrf2. Flavonoids (hesperidin and quercetin), phenols (curcumin and capsaicin), and terpenes (astaxanthin and lutei) are the major dietary modulators of Nrf2 [81]. Low concentrations of epigallocatechin-3-gallate, the catechin most present in green tea, can induce HO-1 through the ARE/Nrf2 cascade in hippocampal neurons, thereby protecting them against various models of oxidative damage [90]. Similarly, caffeic acid phenethyl ester and ethyl ferulate can protect neurons by inducing HO-1 [91].

One of the most intriguing concepts is that natural compounds, such as phytoestrogens, which are structurally similar to estrogens, can interact with both estrogen receptors, ERα and ERβ, and exhibit weak estrogenic activity [92]. A recent study revealed an epistatic link between the Nrf2-Keap1 cascade and steroid hormone-induced signal transduction, demonstrating the function of the Nrf2-Keap1 pathway in neuronal remodeling through an antioxidant-independent and proteasome-dependent activity [93]. This evidence suggests a close correlation within the hormone–nutrition–stress response axis, mediated by a molecular pathway whose alteration can be associated with various pathologies. Hormones and energy metabolism are closely linked, and alterations in energy metabolism can disrupt mitochondrial homeostasis by affecting stress response systems. In this context, endocrine disruptors can also interfere by altering the hormone homeostasis of target organs. First, they bind hormone receptors and modulate their signaling cascades. Endocrine disruptors can alter sex hormones, thyroid hormones, and insulin [94]. Several research publications underline the key role of Nrf2 in enhancing the thyroid antioxidant defense by inducing the expression of cytoprotective factors that play a key role in modulating the normal thyroid function. These factors include GPx2, GR1, thioredoxin 1, thioredoxin reductase 1, sulfiredoxin 1, and NAD(P)H quinone dehydrogenase (NQO1). Studies have recognized irregularities affecting the reproductive and metabolic systems in animal models subjected to endocrine disruptors. Endocrine disruptors can exert a deleterious effect in ocular diseases. Indeed, cataracts, dry eye, macular degeneration, and diabetic retinopathy occur frequently as a consequence of hormone imbalances [94]. The correlation between many retinopathies and sex has been associated with protective effects of gonadal hormones [95]. Gender differences for retinal diseases have also recently been highlighted [96]; the incidence of central serous chorioretinopathy (CSC) has been found to be higher in young adult males [97]. Notably, retinal diseases are characterized by an overlap of neuronal and endocrine alterations together with aging-induced chronic dysfunction. Thus, retinal diseases deserve great attention for the study of the synergic effect of those alterations. 

Despite the fact that the involvement of hormones in the deregulation of molecular pathways is not yet clear, it is known that the Nrf2 pathway can also be subjected to hormonal regulation. Indeed, membrane-associated estrogen receptors (ER)-α36 and G protein-coupled estrogen receptor (GPER) exert an essential function in the estrogen’s fast non-genomic activity, such as the induction of cell proliferation [98]. Through these receptors, estrogen promotes fast Nrf2 induction, modulating the metabolic reprogramming in order to enhance cell proliferation, highlighting the effect of estrogen and phytoestrogens in inducing fast Nrf2 activation through membrane-associated estrogen receptors [98]. A study suggested that silibinin, a compound belonging to the flavonolignan family, induces Nrf2-antioxidative pathways in pancreatic β-cells by modulating ERα expression [99]. High glucose and palmitate induce glucolipotoxicity by decreasing the rat pancreatic β-cell line INS-1 viability, whereas preincubation with 5 or 10 μM of silibinin significantly promoted cell viability. In addition, treatment with ERα-selective agonist 4,4′,4″-(4-propyl-[1H]-pyrazole-1,3,5-triyl)trisphenol (PPT) and ERα-selective antagonist methyl-piperidino-pyrazole (MPP) induced an increase and a decrease in the viability of INS-1 cells, respectively. The anti-inflammatory role of estrogen through the Nrf2 pathway was demonstrated, and it was also reported that E2 induced downregulation of proinflammatory protein expression to a much greater extent in wild type mouse embryonic fibroblast MEFs than in Nrf2 knockout (KO) mice [99]. 

RV can promote epigenetic regulation by inducing the demethylation of the Nrf2 promoter, which was associated with the chemoprotective properties against estrogen-induced breast cancer, which activates the downstream antioxidant genes [100]. The ERβ receptor has also been reported to be linked by S-equol, a gut bacterial metabolite of soy daidzein, inhibiting the interaction of Nrf2 with Keap1 [101]. In support of this mechanism, the induction of estrogen receptor and Nrf2/ARE signaling cascade was also reported by Zhang and colleagues, showing that that S-equol counteracted peroxide-induced endothelial cell apoptosis [102]. Finally, similar molecular pathways involving ERβ and Nrf2 were reported, demonstrating an increase in the expression of the xenobiotic metabolizing enzyme quinone reductase after racemic equol. The complex network that harnesses energy metabolism and food intake in relation to diet and the consumption of bioactive compounds is interesting but at the same time complicated in interpreting the stress mechanisms associated with pathologies closely linked to metabolic disorders. The Nrf2 pathway intersects well with mitochondrial metabolism and with compounds chemically close to estrogen which could reveal interesting perspectives. The challenge to decode this network still remains open [101].

**Figure 4 nutrients-16-03169-f004:**
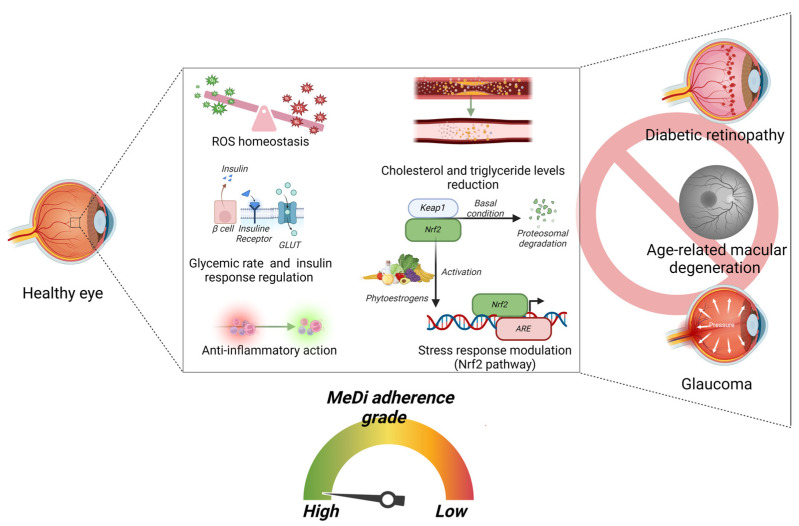
**Pathways involved in maintaining a healthy retina and associated with a high adherence to the MeDi.** Schematic representation of the conditions and signaling pathways that are induced by a high adherence to the MeDi and that participate in maintaining a healthy retina, lowering the risk of DR, AMD, and glaucoma. High adherence to the MeDi results in: (i) a good glycemic control and insulin response; (ii) anti-inflammatory response; (iii) good ROS homeostasis. Moreover, high adherence to the MeDi promotes: (i) reduction in the cholesterol and triglyceride levels; (ii) vegetable/fruit-derived phytoestrogens that activate the Nrf2 pathway, which activates the stress response. All these conditions and pathways induced by a high adherence to the MeDi participate in lowering the risk of DR, AMD, and glaucoma (Created with BioRender.com, Licensing and Agreement number PK2769UG6I).

## 7. MeDi’s Role in the Prevention and Amelioration of Diabetic Retinopathy

Diabetic retinopathy (DR) is a severe diabetes complication characterized by a progressive, chronic, and irreversible visual impairment due to microvascular abnormalities. DR is one of the major causes of blindness in adults worldwide [103]. Every diabetic patient shows a high risk of developing DR (Figure 5).

DR can be classified in different stages of severity, depending on the morphology and functionality of the retinal vasculature. Two types of DR have been defined: non-proliferative diabetic retinopathy (NPDR) and proliferative diabetic retinopathy (PDR) [104]. Hyperglycemia releases inflammatory responses within the retinal environment that initiate the activation, adhesion, and infiltration of leukocytes, followed by the overexpression of inflammatory cytokines [16,17]. The characteristic of NPDR is the presence of vascular aberrations such as microaneurysms and hemorrhages. NPDR can be classified into mild, moderate, or severe stages based on the presence or absence of retinal bleeding. Patients with NPDR generally show hemorrhages of varying sizes, microaneurysms, exudates, and intra-retinal microvascular abnormalities. PDR is a developed stage of DR and characterized by retinal neovascularization due to diabetes-induced ischemia. PDR presents weak vessels that are prone to bleeding, leading to severe vision loss and even blindness [105]. PDR progression induces serious complications including macular edema, retinal detachment, vitreous hemorrhage, neovascular glaucoma, and irreversible blindness [106].

Many studies strongly suggest that diabetes may be prevented with lifestyle changes. Indeed, diabetes can be delayed or prevented by nutritional intervention based on consuming a low-carbohydrate diet, balanced meals, and eating carbohydrates mostly early in the day. Moreover, during a meal, it is better to first consume protein and vegetables and carbohydrates 30 min later, in order to lower glucose levels. Strong evidence has demonstrated that diabetes can be prevented through energy-restricted diets with routine physical activity. MeDi is considered the best dietary pattern for the prevention of diabetes and has received great attention given its role in improving health and reducing the burden of healthcare costs. MeDi reduces the incidence of diabetic retinopathy for type 2 diabetes patients. 

DR development is influenced by hyperglycemia [107] through several pathways: non-enzymatic protein glycation (formation of advanced glycation endproducts (AGEs)), protein kinase C activation, polyol pathway, induction of the hexosamine pathway, accumulation of reactive oxygen species (ROS), and activation of hypoxia-induced factor [108].

MeDi is characterized by whole, nutrient-dense foods and a limited amount of processed and refined foods, which have high sugar content, artificial ingredients, refined carbohydrates, and trans fats. 

Compelling evidence has suggested that the risk of DR can be reduced with diet, demonstrating the protective effect of MeDi. Many studies analyzed the effectiveness of MeDi on the incidence of DR. MD is also considered a beneficial diet for type 1 diabetes mellitus (T1DM) patients [109].

The role of diabetes mellitus (DM) and its progression on the development of DR has been widely analyzed, showing that the amelioration of DM and the maintenance of good glycemic control are beneficial against DR by delaying the onset and slowing the progression of DR. Thus, the dietary intervention mostly acts on the prevention and amelioration of DM and in turn is also beneficial for the DR.

Diaz-Lopez and colleagues demonstrated that supplementation with extra virgin olive oil together with a high adherence to MeDi in more than 3600 participants in a prospective 6-year study reduced the risk of DR (40%). The same study revealed that nut oil supplementation resulted in a low and not significant decrease in the risk of developing DR. The composition of olive oil differs depending on the cultivar, altitude, time of harvest, and extraction process. It contains mainly oleic acid, with other fatty acids such as linoleic acid (up to 21%) and palmitic acid (up to 20%). Oleic acid is a monounsaturated fatty acid that decreases the levels of total cholesterol and low-density lipoprotein. Linoleic acid and palmitic acid are polyunsaturated and saturated fatty acids, respectively. The same study revealed that high intake of fruits and vegetables resulted in a reduced risk of DR, suggesting the relevance of flavonoids in inhibiting the molecular pathways involved in DR.

A clinical trial observed that in subjects with type 2 diabetes mellitus (T2DM), the consumption of at least 500 mg/d of dietary LCω3PUFA, easily obtained with two servings/week of oily fish, correlated with a diminished risk of DR [110]. 

In certain versions of MeDi, milk is substituted with yogurt, kefir, buttermilk, and feta and cottage cheese. Ibsen and colleagues proposed that the substitution of whole-fat yogurt instead of milk among those aged 56–59 lowers the risk of type 2 diabetes, and the replacement of skimmed milk with semi-skimmed milk enhanced the risk among subjects aged 60–64 and 65–72. 

The pathogenic progression of DR can be reduced by specific components that are abundant in MeDi. Indeed, polyphenols that are present in several vegetables, seeds, and fruits decrease insulin resistance and secretion, inflammation, and oxidative stress [111].

Díaz-López and colleagues demonstrated a correlation between lower risk of DR and the intake of flavonoid-rich vegetables and fruits [112]. Diabetic retinal microvascular alteration is clinically characterized by microaneurysms, hemorrhages, lipid exudates, and macular edema in T1DM and T2DM patients. High adherence to MeDi resulted in a decreased risk of retinal microvascular dysfunction. Low intake of fibers correlated with a 41% increase in DR in T2DM patients, compared to T2DM patients more adherent to a high-fiber diet. The PREDIMED study included 7447 Spanish participants and was randomized into three groups: one group consumed a highly adherent MeDi supplemented with extra virgin olive oil, the second followed a highly adherent MeDi enriched with mixed nuts, and the control group was subjected to a low-fat diet for a median of 5 years. This Spanish cohort study showed that MeDi enriched in extra virgin olive oil or nut intake significantly decreased the incidence of major cardiovascular events compared to a low-fat diet. However, only MeDi enriched with olive oil protected against DR (60% decrease in DR), while MeDi enriched with nuts resulted in a 37% decrease in DR [112]. In middle-aged and aged T2DM patients, consumption of at least 500 mg/d of dietary LCω3PUFA, obtained with two servings/week of oily fish, correlated with a diminished risk of sight-threatening DR [113]. 

Oleic acid is an essential component of olive oil. It is a monounsaturated fatty acid. Olive oil shows the presence of polyphenols and vitamins K and E, which can lower oxidative stress, inflammation, and insulin resistance [114]. 

Nuts contain RV, a polyphenolic compound contained in several plants such as grapes and peanuts, which plays a role in promoting anti-obesity, cardioprotective, neuroprotective, antitumor, antidiabetic, antioxidant, and anti-aging effects, and modulates glucose metabolism. The effects of RV are modulated by various synergistic pathways converging in the control of oxidative stress, cell death, and inflammation. RV modulated apelin gene expression in a rat model of T2DM [115]. The authors found a significant decrease in serum glucose level in rats treated with 5 and 10 mg/kg per day with RV compared with the diabetic control. In agreement, resistin expression in adipose tissue was reduced in RV-treated groups. RV induced heme oxygenase-1 (HO-1) expression through ARE-mediated transcriptional activation of Nrf2, suggesting that RV augmented cellular antioxidant defense capability following the induction of HO-1 through Nrf2-ARE cascade [116]. RV supplementation in diabetic rats resulted in significant amelioration of hyperglycemia, weight loss, increased oxidative markers, superoxide dismutase activity, and inhibition of eNOS activity in the blood and retina [117]. 

It has been shown that intake of at least 500 mg/d of dietary LCω3PUFA correlated with a decreased risk of sight-threatening DR [113]. Fish intake at least twice a week correlated with a 60% reduction in DR risk [108]. The anti-inflammatory effect of omega-3-fatty acid exerts an essential function in reducing the risk of DR [118]. Fish consumption in Japan, five times higher than in Western countries, reduces the incidence and progression rate of diabetic retinopathy compared to Western populations [119]. 

Various studies revealed the essential role of micro- and macro-elements in DR. Brazionis and colleagues indicated that plasma carotenoid levels seem to play a role in diabetic retinopathy, independent of established risk factors [120]. Lutein supplementation was shown to delay DR progression within 5 years according to Garcia Medina and colleagues [121]. 

Tanaka and colleagues demonstrated that fruit intake correlated with a decreased incidence of diabetic retinopathy among patients following a low-fat energy-restricted diet [122]. Fruits are low-glycemic-index foods enriched in fibers that can slow glucose response. Several investigations indicate that adherence to MeDi and high fruit intake are beneficial against the development of diabetic retinopathy. Post and colleagues suggested that fiber supplementation T2DM patients lowered fasting blood glucose and HbA1c [123]. Supplementation with vitamin C diminished the risk of retinopathy [122]. High vitamin C consumption correlated with 40% decreased risk of retinopathy [122] and the intake of vitamin C together with statins diminished the effects of non-proliferative DR more than statins alone [124]. 

Vitamins have a beneficial effect in lowering the risk of DR. Chatziralli and colleagues reported that vitamin E reduced serum malondialdehyde levels and oxidative stress, suggesting that vitamin E supplementation produced an additional benefit by lowering the risks of developing DR progression [125]. Oral vitamin E treatment (1800 IU daily vitamin E) was effective in normalizing retinal hemodynamic abnormalities and enhancing renal function in T1DM patients [126]. Vitamin C protected against diabetic retinopathy progression. Vitamin C exerts antioxidant and anti-angiogenic actions and enhanced endothelial function [127]. Barba and colleagues showed that PDR patients showed decreased intra-vitreous concentrations of ascorbic acid compared to non-diabetic patients [128]. Vitamin C is an anti-oxidant and modulates oxygen tension and eye-oxidative stress. The lower level of intra-vitreous content of vitamin C is caused by competition between the glucose and the ascorbic acid to bond the GLUT-1 glucose transporter. Rafael Simó and Cristina Hernández suggested that regular intake of foods enriched in vitamin C, including citrus fruits, together with good glycemic control, was important for preserving the correct intra-retinal levels of ascorbic acid [127]. Park and colleagues observed that the vitreous level of vitamin C in PDR patients were diminished tenfold, which correlated with the degree of macular ischemia, suggesting that vitreous vitamin C depletion can promote macula ischemia in PDR patients [129]. The association of vitamins E and C enhanced the antioxidant effectiveness in the retina [130]. Other vitamins, including vitamins D and B, can exert a pivotal function in decreasing DR risk. The correlation between vitamin D deficiency and retinopathy severity was found in diabetic patients with well-controlled glycemia, suggesting the function of vitamin D in reducing the risk and severity of DR [130]. Vitamin B6 is beneficial against the early death of pericyte cells by maintaining the viability of capillaries, helping to maintain the presence of microvascular cells [114]. 

Others compounds can decrease the development and progression of DR, like zinc, iron, and manganese copper [114].

**Figure 5 nutrients-16-03169-f005:**
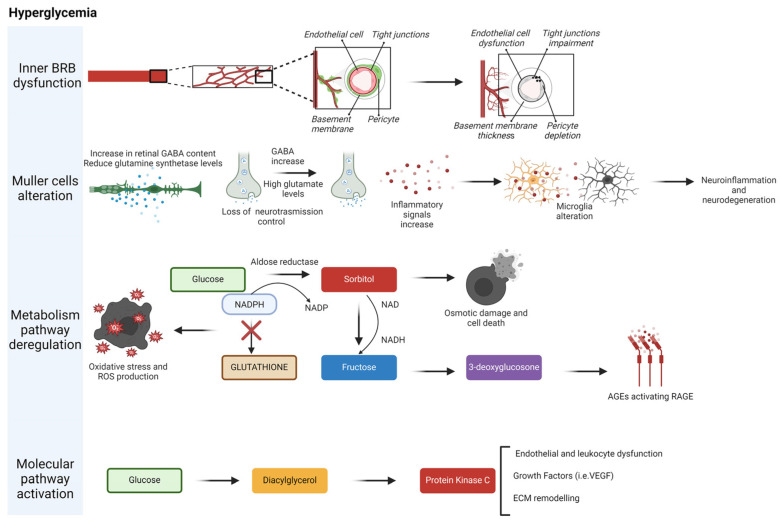
**Cellular, metabolic, and signaling alterations in diabetic retinopathy.** Four main interconnected mechanisms in retinopathy are shown. Müller cells have numerous functions, including maintaining the proper functioning of the blood–retina barrier. Additionally, they are involved in regulating synaptic neurotransmission and providing neuroprotection. Finally, the most affected processes involve metabolic and genomic pathways linked to hyperglycemia, such as the polyol pathway and the activation of protein kinase C. Created in BioRender.

## 8. MeDi’s Role in the Prevention and Amelioration of Age-Related Macular Degeneration (AMD)

Age-related macular degeneration (AMD) is a retinal degenerative disorder affecting subjects over the age of 55 years. It is a major cause of blindness in industrialized countries, from 196 million affected individuals in 2020 to 288 million by 2040. AMD is a heterogeneous illness, influenced by age, genetics, and environmental factors, such as smoking and diet. Three stages characterize AMD: early, intermediate, and late. The first and second stages, characterized by the absence of symptoms or mild visual symptoms, show macular deposits (drusen) and pigmentary abnormalities [131]. Late AMD can show neovascular and atrophic forms characterized by macular neovascularization and atrophy. The neovascular stage could exist in two subtypes: the non-neovascular (dry) type and the neovascular (wet) type. The most frequent symptoms of age-related macular degeneration are blurry or fuzzy vision, difficulty in recognizing familiar faces, a dark, empty area or blind spot appearing in the center of vision, and the loss of central vision. In the late stage, visual symptoms are serious and irreversible, and include significantly diminished central vision in both eyes. 

Many researchers have shown that eating vegetables, fruits, and fish in a Mediterranean-inspired diet is useful in protecting against age-related macular degeneration (AMD). Hogg and colleagues analyzed the correlation between AMD and MeDi in seven European countries [132]. They found that the populations with the highest MD score have the lowest level of advanced AMD [132]. The association between healthy diet, physical activity, and not smoking correlated with 71% lower chance for AMD [133]. In another study, 41,514 participants aged 40 to 70 years and born in Australia or New Zealand who migrated from the United Kingdom, Italy, Greece, or Malta were recruited and assessed for AMD prevalence in a follow-up study analyzing the effect of dietary habits on the onset and progression of AMD. It was demonstrated that dietary factors can regulate AMD risk. Predominant intake of grains, fish, steamed or boiled chicken, vegetables, and nuts correlated with a lower prevalence of advanced AMD, whereas red meat consumption correlated with a higher prevalence of advanced AMD [134]. Merle and colleagues showed that high adherence to the MeDi correlated with a 41% decreased risk of incident advanced AMD in two European population-based prospective cohorts [135,136]. Advanced AMD risk was lowered by 22%, 26%, and 47% in studies by Keenan [131], Merle [136], and Merle [134], respectively.

Drusen size progression decreased by 17% when following the most adherent MeDi compared to subjects following a less adherent MeDi [135]. Analysis of AMD progression showed a significant correlation between highly adherent MeDi and slower enlargement of atrophy [137].

Merle and colleagues studied the correlation between MeDi, AMD, and genetic susceptibility [136]. High adherence to MeDi correlated with a 26% lower risk of progression to advanced AMD [136]. Consuming fish and vegetables reduced the risk of progression of AMD. Genetic variations between different populations are also associated with AMD prevalence [136]. In particular, the prevalence of AMD in different ethnicities and geographic regions should consider genetic variations, in particular the single-nucleotide polymorphism (SNP) Y402H in the complement factor H (CFH) gene [138]. The risk of AMD progression was also significantly decreased among subjects carrying the *CFH* Y402H allele (T), while the individual homozygous for risk allele (CC) showed an enhanced risk of AMD progression [136,139]. European populations showed higher frequencies of risk alleles, correlating with increased progression of AMD, compared to Chinese and Japanese descendants [140]. The relationship between geographic region and prevalence of AMD could be at least partially explained by gene–diet interaction [141]. However, Hogg and colleagues did not find any correlation between AMD progression, MeDi adherence, and the presence of the Y402H allele, probably because a small number of neovascular AMD cases were analyzed [132]. On the contrary, Keenan and colleagues found that patients carrying the rs10922109 allele showed a lower risk of atrophy compared to neovascular AMD when their diet was highly adherent to MeDi [131]. 

Inflammation markers, including C-reactive protein (CRP), interleukin 6 (IL6), E-selectin, and soluble intercellular adhesion molecule 1 (sICAM-1), exert a function in diabetes development. A positive correlation between serum concentrations of sICAM-1 and E-selectin and diabetes risk has been shown [142]. Notably, the correlation between diet and diabetes is regulated in part through the modulation of the inflammatory response [142], suggesting that the MeDi can modulate the progression of AMD by acting on the inflammatory response.

The beneficial effects of the MeDi are correlated with a decrease in oxidative stress and inflammation, which exert a significant function in AMD [142]. Subjects following a highly adherent MeDi show elevated serum levels of biomarkers considered beneficial against AMD [136]. A high adherence to MeDi is more effective than the consumption of antioxidant and zinc supplementation. Trials examined short- or intermediate-term effects of MeDi on circulating markers of oxidative stress, such as urinary F2-isoprostanes, plasma malondialdehyde, and oxidized LDL. It has been demonstrated that subjects following a highly adherent MeDi showed lower oxidized LDL compared to the control group [143]. Coliij and colleagues proposed that elevated HDL cholesterol levels correlated with augmented risk for AMD [143]. Moreover, it has been shown that healthy habits and a healthy diet associated with supplement assumption are important for the prevention of ADM progression to late stages [136].

Jiang and colleagues demonstrated that a high intake of dietary omega-3 PUFA or fish correlated with a decreased AMD risk [144]. Moreover, a diet enriched in fish and with a low content of linoleic acid reduced the risk of AMD [145]. Interestingly, a high intake of ω-3 fatty acids or fish has no effect in preventing AMD progression in subjects consuming high levels of dietary linoleic acid [145]. Oily fish consumption at least once a week resulted in a lower risk of AMD progression [146]. RV, a bioactive compound present also in nuts, has antioxidant, antithrombotic, and anti-inflammatory properties [147]. It was demonstrated that RV prevented apoptosis of human retinal pigment epithelial (RPE) cells in vitro [148]. Moreover, RV protected RPE cells from autoimmune antibody-promoted apoptosis in vitro [148]. RV prevented oxidative stress-induced RPE degeneration by promoting the activity of superoxide dismutase, glutathione peroxidase, and catalase [149]. Nutritional supplementation with RV exerted beneficial function that resembled the effects induced by anti-VEGF treatment, promoting the anatomical restoration of retinal structure, RPE function, and choroidal blood flow [150]. 

High dietary intake of lutein correlated with a lower risk of prevalence and incidence of AMD. Lutein is present at elevated concentrations in green leafy vegetables such as spinach, kale, and yellow carrots and also in animal fat. Several studies unveiled the beneficial effect of lutein in lowering AMD risk [151,152]. Lutein is a filter for blue light. For this reason, lutein supplementation has been shown to protect the fovea from blue light-induced damage [153]. Moreover, membrane-bound lutein was demonstrated to exert an ROS scavenger function [154]. Indeed, the unconjugated double bonds in the molecular structure of lutein have a function in ROS quenching. Moreover, lutein decreased lipofuscin accumulation in cultured RPE cells by decreasing oxidative stress [155]. Dietary supplementation with lutein and zeaxanthin for 6 months increased the optical density of macular pigment [150,156,157,158]. Dietary lutein and zeaxanthin intake decreased the risk of incident early or neovascular AMD over 5 and 10 years [159]. Moreover, lutein, zeaxanthin, eicosapentaenoic acid, and docosahexaenoic acid, which show elevated levels in MeDi, were associated with diminished serum levels of C-reactive protein, suggesting a function in decreasing systemic inflammation in AMD subjects [160]. 

Vitamins seem to exert a relevant role in lowering AMD risk, since vitamin C is abundant in the retina. Although several studies indicated a correlation between dietary consumption of vitamin C and AMD risk, the function of vitamin C in preventing AMD risk is still controversial. Seddon and colleagues did not find any significant correlation between vitamin C consumption and reduced risk for AMD [161]. In agreement, a more recent study confirmed that vitamin C supplementation did not prevent the risk and the progression of AMD [162]. On the contrary, SanGiovanni and colleagues reported a decreased risk of developing neovascular AMD in subjects consuming elevated levels of dietary β-carotene, vitamin C, and vitamin E [163]. Vitamin E shows an antioxidant activity and RPE show a high concentration of vitamin E, suggesting a protective role of this vitamin in RPE. Vitamin E is composed of four different compounds: α-tocopherol, β-tocopherol, γ-tocopherol, and δ-tocopherol, with α-tocopherol as the most effective scavenger of free radicals. Wiegand and colleagues demonstrated that vitamin E concentrations in the retina were enhanced in response to augmented oxidative stress [164]. Dietary deprivation of vitamin E resulted in an augmented lipofuscin accumulation in the RPE [159]. Moreover, vitamin E deficiency accelerates retinal degenerative damage [165]. Low serum levels of tocopherol were associated with AMD progression [166]. In agreement, an association has been found between fasting α-tocopherol levels and AMD progression to the late stages [167]. Although various studies suggested a protective role of vitamin E in the prevention/amelioration of AMD, other clinical studies did not reveal any significant beneficial effect induced by vitamin E supplementation in AMD prevention or risk reduction [168,169]. Thus, the efficacy of nutrient supplementation in preventing AMD risk is still debated. Although studies show the benefit of specific nutrients, Merle and colleagues demonstrated that none of the nine components of the MeDi, such as vegetables, fruits, legumes, cereals, fish, the MUFA-to-SFA ratio, meat, dairy products, and alcohol intake, significantly correlated with the incidence of advanced AMD, underlining the relevance of assessing dietary patterns rather than single components [134].

## 9. MeDi’s Role in the Prevention and Amelioration of Glaucoma

Worldwide, glaucoma is one of the major causes of irreversible blindness and significant vision impairment due to elevated intraocular pressure (IOP), which is a key modifiable risk factor for preventing the death of retinal ganglion cell [170,171].

To date, the European Glaucoma Society accounts for more than 70 million people with glaucoma diagnosis, with an estimated increase of about 112 million by 2040 [172,173]. It is considered a debilitating and heterogenous neurodegenerative eye disorder and it is characterized by progressive damage in retinal ganglion cells, a bridge between the inner surface of the retina and the optic nerve. Retinal ganglion cell degeneration will lead to the death of smaller nerves around the optic nerve up to a more pronounced blindness [174]. Although peripheral vision is affected first, mainly without the patient realizing it, later stages can destroy in an irreversible way the central visual field [175]. Notably, glaucoma can be divided in two groups, open-angle glaucoma (OAG) and closed-angle glaucoma (CAG), exhibiting characteristic morphological modifications in the optic nerve head and retinal nerve fiber layer. OAG typically occurs with an open drainage angle in the eye, while CAG, which is less common and represents a medical emergency, involves a closed or blocked drainage angle [176]. The disorders of visual function can be related to several risk factors as well as inadequate quantitative and qualitative nutrient supply, onset of genetic alterations, type 2 diabetes, obesity, hypertension, high myopia, environmental factors (pollution, UV rays, cigarette smoking, and particles), old age, and ethnic background (Afro-American or Hispanic) [9,177,178]. Generally, glaucoma is treated with eye drops that decrease the IOP to prevent damage to the optic nerve; however, these treatments, unfortunately, cannot restore the vision loss or cure this disease [179]. For those reasons, the early detection and intervention of modifiable risk factors, such as those linked to MeDi and dietary supplements, could be crucial for reducing the incidence of glaucoma and decelerating its development. Since numerous chronic conditions associated with glaucoma stem from dietary and metabolic disorders, it is clear that nutrition exerts a crucial function in the development, prevention, and treatment of this illness.

However, healthy foods, including fruits, whole grains, olive oil, vegetables, seafood nuts, and beans, which are included in MeDi, have been linked to the prevention of chronic age-related diseases (AREDs). In this context, a systematic review suggested that the consumption of a defined dietary pattern, when compared with a single component or nutrient, could imply potential protective effects by lowering the incidence of OAG (iOAG), although such evidence should be better explored in more detail [180,181]. Thus, in one of the most recent case–control studies, the Rotterdam Study, Vergroesen JE and colleagues demonstrated the correlation between the adherence to Mediterranean-DASH Intervention for Neurodegenerative Delay (MIND) and reduced risk of iOAG. In three independent cohorts from the prospective population, 170 participants developed iOAG in 1991 with follow-up visits every five years. In this program, a high consumption of food rich in nutrients and fiber and low in calories and fat, as well as seafood, strawberries and blueberries kale, collard greens, spinach, cabbage, and so on, with the latter included in green leafy vegetables, showed both anti-inflammatory and protective activities against iOAG. Moreover, the authors also investigated a possible adherence to MeDi or other guidelines, as well as a Dutch diet, and iOAG, but they did not find any remarkable associations. Considering that an IOP-independent correlation was demonstrated, the authors concluded that the MIND diet tended to be effective in slowing down or halting the progressive neurodegeneration of the optical nerve [180]. Another interesting study came from Moreno-Montañés and colleagues, in a large prospective cohort with more than 10 years of follow-up time (updated with self-reported questionnaires that included lifestyle changes, health-related activities, and medical interventions). In the “Seguimiento Universidad de Navarra” (SUN) Project, the authors assessed the impact of the Mediterranean lifestyle (ML) habits (among no history of smoking, moderate and/or high physical activities, MeDi adherence, body mass index, modest alcohol consumption, and working 40 h per week, to which corresponds the SUN Healthy Lifestyle Score, SHLS, to define the adherence) on the risk of developing glaucoma. As a result, 261 (1.42%) new cases of glaucoma were diagnosed in the largest cohort ever reported with a total of 18,420 participants. They observed a decreased risk of glaucoma in participants with higher SHSL scores (>6) which adhered better to ML, while no significant association was demonstrated regarding MeDi related to each of its components. This work outlined, for the first time, that ML may reduce the incidence of glaucoma as a modifiable and protective risk factor, with a healthy lifestyle system [182]. Although the link between ML and glaucoma remains to be ascertained, a possible explanation could be attributable to the alterations in the nitric oxide (NO)–guanylate cyclase (GC) pathway. As recently reported, MeDi provides L-arginine and nitrate, which act as NO precursors, as well as vitamins, polyphenols, and fatty acids, which potentially boost NO endogenous production, providing both anti-inflammatory and anti-apoptotic properties [183,184,185]. A previous prospective analysis from the Nurses’ Health and the Health Professionals Follow-up Study (63,893 women and 41,094 men, respectively) was reported by Kang and collaborators, who demonstrated that both higher total dietary nitrates, as an exogenous NO source, and vegetable intake were associated with lower IOP and risk of OAG and its subsequent progression. The reason could be due to the elevated concentrations of antioxidants and flavonoids present in these foods, which exerted neuroprotective effects [186]. According to the last piece evidence, Abreu-Reyes and colleagues performed an observational study, then validated it with the Prevention through Mediterranean Diet (PREDIMED), on 100 Spanish Canary Islands patients with the diagnosis of OAG in terms of their adherence to MeDi. Briefly, the authors reported only moderate adherence to MeDi with a high % of participants, approximately 70%, without gender differences [187]. Recently, an extensive review provided by Valero-Vello M and collaborators focused on nutritional hallmarks of foods and oral supplements in a Mediterranean cohort. As a consequence, they did not find any significant correlation between the adherence to MeDi by age and/or gender and the restoration of optic nerve damage in glaucoma patients. Overall, since MeDi plays an important preventive role against progressive eye conditions, the combination with nutritional supplementations as adjuvant factors would allow for high the adherence to healthy diet patterns, thus preventing the vision loss and increasing the quality of life of glaucoma patients [111]. Moreover, Mvitu and colleagues carried out a cross-sectional study that counted 244 Congolese patients affected by type 2 diabetes mellitus (T2 DM, 48% of males; 40% aged ≥ 60 years). The assessment of dietary intake was linked to a qualitative-type questionnaire that resumed the frequency of red beans, vegetable, fruit intake, and cataract extraction. Interestingly, they noticed that regular MeDi intake (*Abelmoschus*, *Brassica rapa*, *Musa acuminate*, beans) decreased the risk of blindness, cataracts, and glaucoma in this group of patients; nevertheless, these results focused on a very low rate of vegetable intake in Africa. Particularly, from univariate analysis, red bean intake and consumption equal to or more than three servings of vegetables per day represented independent and protective factors against eye degeneration diseases [188]. To date, further high-quality studies are required to deeply elucidate both molecular mechanisms and healthy benefits of MeDi in the prevention of glaucoma [181]. 

A large amount of evidence suggests that the effects of nutritional supplements positively impact several ocular dysfunctions, acting as a powerful neuroprotective on the modulation of IOP in preclinical animal models and patients affected by glaucoma [189,190]. For this purpose, a systematic review reported that a high dietary consumption of some micronutrients derived from leafy green vegetables like kale and spinach that are rich in vitamins, minerals and fibers, and contain for instance flavonoids, glutathione and NO, led to reduced levels of reactive oxygen species (ROS) and, consequently, the risk of glaucoma onset in patients affected by OAG. These findings were different from selenium (Se) and iron (Fe), contained in red meats, which would seem to increase the risk of developing glaucoma, although randomized clinical trials (RCTs) will be necessary to confirm these results [191]. Other supplementations as well as blackcurrant, an optimal source of polyphenols, provided a significant improvement in the visual field and ocular blood flow (OBF) in 38 OAG patients, during 24 months of follow-up, but on the other hand, no effects were observed in IOP changes [192]. In addition, Lee J. and colleagues studied the long-term effects on the visual field, particularly at the superior central, following the supplementation of ginkgo biloba extracts, over 12 years of follow-up in a group of 42 patients with normal tension glaucoma (NTG), demonstrating a slower progression in the evaluation of global indices [193]. Alternatively, the supplementation of omega-3 fatty acids, reported by Garcia-Medina et al., did not show any effective treatment in 117 subjects with mild or moderate POAG and IOP over a 2-year follow-up period [194]. 

Another study considered a possible association between nutritional supplements as well as supplementation with eicosapentaenoic acid (EPA) and docosahexaenoic acid (DHA), two polyunsaturated fatty acids that are typically found in fatty fish, fish oil, and algae, on age-related macular degeneration (AMD) at intermediate and/or advanced stage in a prospective cohort from the Nurses’ Health and the Health Professionals Follow-up Study (75,889 women and 38,961 men, respectively). The authors found that an increase in intake of EPA and DHA could slow down the development of visual dysfunction in the intermediate AMD stage [68]. These results confirmed some beneficial effects obtained by previous double-blind, placebo-controlled studies analyzing the anti-inflammatory and neuroprotective effects of EPA and DHA in glaucoma treatment in a dose-dependent manner. Moreover, the study demonstrated that these effects are mediated by peroxisome proliferator-activated receptors PPAR-α, PPAR-γ and PPAR-δ [195]. Further healthy benefits about nutritional supplements, particularly from saffron, came from two studies to evaluate IOP reduction in 22 and 34 OAG patient cohorts with a short-term follow-up, respectively. The first one published by Hecht and colleagues did not show any hypotensive effect deriving from a supplementation of 1g twice/week of saffron to OAG patients [196], whereas the second study subministered 30 mg/day of aqueous saffron dose, after three weeks of conventional timolol and dorzolamide therapy. In these conditions, an ocular hypotensive effect was evident, confirming the anti-inflammatory and neuroprotective role of saffron against glaucomatous optic neuropathy [197]. These results supported the data obtained in vitro by a study reported by Fernández-Albarral JA and colleagues in a mouse model of chronic ocular hypertension (OHT) [198]. Extra virgin olive oil (EVOO) induces beneficial effects because of both anti-inflammatory and antioxidant properties that are provided by the presence in EVOO of over 30 phenolic compounds, as well as oleuropein, verbascoside, tyrosol, hydroxytyrosol, diosmetin, luteolin and rutin, which counteract the pathological pathways that participate to de progression of glaucomatous degeneration [199]. Notably, the first two aforementioned bioactive compounds, oleuropein and verbascoside, exhibited a significant inhibitory effect at low μM concentrations in vitro against human carbonic anhydrase I and II (hCA I and II) isoenzymes [200], which are therapeutic targets against glaucoma and their inhibition is considered a therapeutic strategy against glaucoma [201]. The enzymatic characterization of natural phenolic compounds as well as flavonoids was useful to obtain a clinical amelioration in visual function, minimizing the risk of ophthalmic artery occlusions for those patients with glaucoma [202]. To investigate whether the impact of high dietary fat and sucrose in animal models was able to induce the injury of retinal ganglion cells (RGCs), Chrysostomou and collaborators demonstrated that C57BL/6J mice fed with a short-term high fat/high sucrose diet were more vulnerable to optic nerve damage and showed higher intraocular pressure following the injection of endotoxin-free saline [203]. Similarly, Kong and collaborators tested the effect of diet restriction (DR, with alternate- fasting plan at least for 6 months) in older (18-month) C57BL/6J mice with an inner retinal dysfunction during and after injury caused by IOP. DR treatment resulted in an appreciable functional recovery of retinal neurons at the inner level and enhanced the mitochondrial activity in the retina of older animals when compared with age-matched control mice. The author found that DR decreased ROS levels and oxidation products as well as the levels of oxidative stress markers (heme oxygenase-1, HO-1, and 4 hydroxynonenal, 4-HNE) compared to IOP mice fed with a normal diet [204]. Additionally, Guo X and colleagues tested an every-other-day fasting (EODF)—a form of caloric restriction-, to assess its effects on glaucomatous pathology in EAAC1-/- mice, an animal model with a normal tension glaucoma. They showed that EODF exhibited a neuroprotective function with an improvement of visual impairment in these mice models by ameliorating RGCs and retinal degeneration without modifying IOP [205]. Although the molecular mechanisms induced by dietary restrictions are not yet entirely elucidated, the observed neuroprotective effects seem to be associated with the induction of autophagy and the improvement of retinal ganglion cells survival, indicating that this cytoprotective process could represent an useful therapeutic strategy in glaucoma following the exposure to hypoxic/ischemic stress [206,207]. Surprisingly, a direct correlation was found between hypertensive patients with elevated IOP and higher levels of melatonin in their aqueous humor compared to the normotensive group. This correlation was previously reported in the experimental glaucomatous model (DBA/2J) compared to control mice (C57BL/6J). The authors speculated that the increase of melatonin in the humor was due to hyperactivation of the transient receptor potential vanilloid-type 4 (TRPV4) cation channel that induced higher melatonin levels. These data suggested that IOP promoted an antioxidant protective response by enhancing melatonin concentrations [208]. Indeed, Melatonin exerts a beneficial effect against glaucoma by blocking the oxidative stress-promoted degeneration of the retinal ganglion cells and is proposed as a therapeutic strategy against glaucoma [209,210,211]. Further evidence about the association between melatonin levels in both aqueous humor and serum and eye disorders in type 2 diabetic patients derived from the study reported by Aydin and colleagues. They hypothesized that the increase in melatonin levels in the eye of glaucoma patients could be due to intraocular concentration and not by melatonin from the pineal gland [212]. In summary, although a large number of studies regarding the long-term advantages and safety of supplements in glaucoma patients seems to be variable, the possibility to find a useful and synergistic combination of different antioxidants or bioactive compounds against sight-threatening or lifelong diseases could be a promising therapeutic option [213]. Moreover, the beneficial effect of MeDi in the prevention of glaucoma is still debated [181].

## 10. Conclusions

Certainly, healthy life habits prevent the onset and the progression of chronic diseases. Concerning DR, AMD, and glaucoma, several studies underlined the relevance of MeDi for the prevention of these ocular diseases, mostly by preventing oxidative damage and chronic inflammation. Notably, the studies focusing on the effects of supplements are still controversial, suggesting that the adherence to MeDi and the Mediterranean lifestyle exert a major effect in the prevention of retinal diseases and that supplements can be an adjuvant to MeDi, but cannot substitute for healthy dietary habits in the prevention of such diseases. In addition, the studies focusing on the efficacy of MeDi also provide some controversial results, probably due to the different methods used for calculating MeDi scoring. Notably, such studies directly correlate MeDi adherence with the onset and progression of retinal diseases, but there is not a biomarker that clearly associates the effect of the dietary pattern to the disease progression. The absence of a diet-induced or repressed biomarker is a major problem in correlating the effects of dietary patterns with the risk of developing retinal diseases. Herein, we propose to investigate in more detail the effect of MeDi in lowering the serum levels of AGEs and in promoting the activity/expression of key molecules involved in the Nrf2 pathways, in order to have a clear molecular target that will provide a defined measure of the efficacy of MeDi and a molecular correlation with the progression of retinal diseases.

## Figures and Tables

**Figure 1 nutrients-16-03169-f001:**
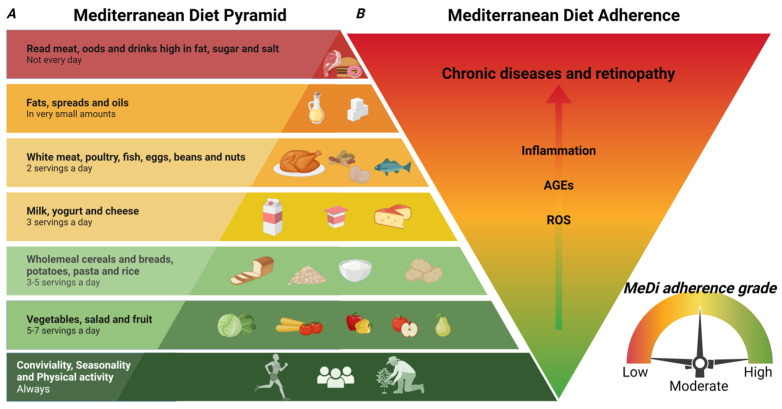
(**A**) **The MeDi pyramid.** The basis of the pyramid includes the Mediterranean lifestyle, with conviviality and daily moderate physical activity. The food categories and the frequency of consumption that represent a high adherence to the MeDi are indicated. (**B**) **MeDi adherence.** Progressive effects induced by lowering the adherence to the MeDi from high (green) to very low (red): starting from ROS production, to increased AGEs formation, followed by inflammation and finally chronic progression of retinopathy (Created with BioRender.com, Licensing and Agreement number GT2769TZYW).

**Figure 2 nutrients-16-03169-f002:**
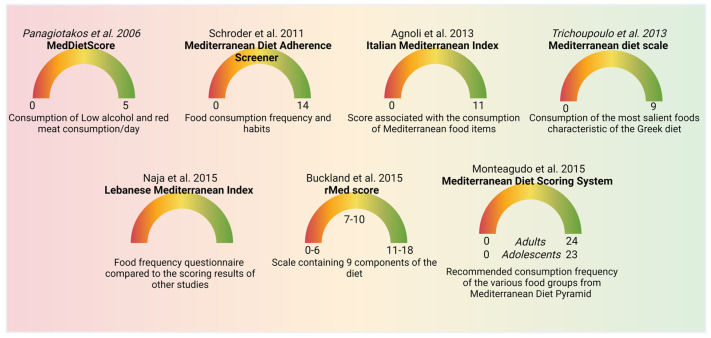
**Schematic representation of the various MeDi scoring methods.** The different MeDi scoring methods are indicated together with the adherence to the MeDi from lower to higher score (Created with BioRender.com, Licensing and Agreement number SX2769U3H2) [25,26,27,28,29,30,31].

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
