# Peer review of "Effect of the Mediterranean Diet (MeDi) on the Progression of Retinal Disease: A Narrative Review"

_nutrients, 2024, doi:10.3390/nu16183169_

Round 1

Reviewer 1 Report

Comments and Suggestions for Authors

Here authors review the effects of the Mediterranean Diet (MeDi) in the progression of retinal diseases. This is an important topic in terms of quality of life as hundreds of millions of humans face different visual impairments worldwide. As the topic is vast, there are several points that need attention so the manuscript can improve

1.        In Abstract Line 10 authors mention 1 billion affected by visual impairment as well as blindness; In line 41, 600 million, while in line 43 - 543 million - please assure an accurate number of patients worldwide affected with the correct reference.

2.        Lines 62-71 - Authors must develop a better/deeper paragraph related to the highly ordered architecture of the retina containing photoreceptors, which are targeted by several diseases, interneurons, projection neurons (Retinal Ganglion Cell, a target as well, as the optic nerve), and three types of glial cells. Please, describe if the MeDi can ameliorate the structure/function of cell bodies of retinal cells organized in three nuclear layers, cell processes and synapses (Glutamate transporters that avoid excitotoxicity). As the manuscript deals with the retina, and hundreds of millions are affected by visual problems, a schematic figure could illustrate which cells are targeted in different diseases)

3.        Lines 73-100 – Mediterranean diet is defined by: whole grains, olive oil, fruits, vegetables, seafood, beans and nuts, which are included in the MeDi, have been linked to the prevention of chronic age-related diseases (AREDs). Are these all the constituents of MeDi? Should a moderate glass of wine be included? Please describe what is in common and what differs from all types of MeDi (Lebanese, Italian and others) in terms of constituents and then jump to the pyramid line 90

4.        Lines 111-181 – Authors show several important data in a not clear manner… These differences in the scoring result in an increased difficulty in comparing the data” (I agree). Please try to illustrate information (that are essential to the topic chosen) so the reader finds easier to compare – please use table; differences in terms of quality and healthy benefits of different MeDi (Italian, Lebanese, and others)?  

5.        Lines 167-171 reads...In adults, 1 point is added for alcohol intake equivalent to 1 (women) and 2 (men) glasses of wine or beer. Sofi and colleagues developed the MeDi-Lite score in 2017 that analyzed 9 food groups and compared the results with the MDS score, demonstrating that the higher range score provided an increased sensibility and specificity compared to MDS scoring [21]. How can one evaluate the level of resveratrol, alcohol content and the type of grape for the wine added in the scale? For example, Pinot-noir, Malbec, or Carmenere, any difference? Vitis vinifera, labrusca and muscadine types of grapes contain high concentrations of resveratrol (50-100ug/g), but how one can relate to the type of wine chosen?

6.        Lines 258-270 Linoleic acid (LA) and alpha-linoleic acid (LNA) are essential polyunsaturated fatty acid (PUFA) obtained from several food categories listed in this manuscript (sunflower oil, grape seed oil, safflower oil, walnuts, salmon, chia seeds, sardine, and others 

7.        …Several studies demonstrate that the ratio between omega-3 and omega-6 fatty acids is important for the prevention of cardiovascular diseases. Important to mention that LA and LNA are essential lipids obtained in diet through intake of vegetable and animal derived oils…. PLEASE COMMENT ON THE BIOCHEMICAL ROUTES AFTER CONSUMPTION, n-3 and n-6 fatty acids directed to separate pathways for β-oxidation, storage, or elongation. Both LA and LNA undergo several steps of desaturation (Δ6 and Δ5 desaturases) and elongation, generating a great number of metabolites, including ARA, from LA, and EPA and docosahexaenoic (DHA) acids, from LNA. ARA generates the commonly known ECs anandamide (AEA) and 2-AG, while eicosapentaenoyl ethanolamine (EPEA) and docosahexaenoyl ethanolamine (DHEA) are produced from EPA and DHA, respectively, recently recognized as weaker ECs. 

Please mention that DHEA is also known as synaptamide -a trophic factor that improves cognitive parameters in the nervous system. It is a metabolite generated on the omega-3 arm

8.        Lines 271-275 Dietary intake of omega-3 fatty acids by consumption of fatty fish (IS THIS CORRECT?) or fish oil exerts a beneficial effect by decreasing the risk of cardiovascular diseases, inflammatory bowel disease, cancer, rheumatoid arthritis, psychiatric and neurodegenerative diseases [47]. Omega-6 PUFA is present in extra virgin olive oil. PUFA are contained also in nuts, at high level in pistachio. There are several types of oils that were studied in terms of omega-3/omega-6 contents that could be cited. Please see Iorsavova et al., Int J. Mol. Sci 2015

9.        Line 294 Saffron is a medicinal plant that has strong anti-inflammatory and antioxidant action, having several health benefits, such as relieving PMS symptoms, helping to control diabetes, promoting weight loss and preventing cardiovascular diseases, for example. Several of the nutraceuticals listed in the manuscript (resveratrol and others found in (1) turmeric root; (2) grapes; (3) green tea; (4) Maytenus senegalensis (Lam.) fruit and roots; (5) black peppers; and (6) blueberries modulate cannabinoid CB1 and/or CB2 receptors, interacting with the endocannabinoid system, enzymes or receptors. Therefore, mechanisms can be explained Curcumin that is present in Saffron and is listed as well.

10.  Figure 3 - retinal diseases might show complete different set of mechanisms - glaucoma, retinitis pigmentosa, age macular disease, diabethic retinopathy...Authors should specify mechanisms in how the elements/constituents/active principles of MeDi might improve different functions in the retina. I could list several that are important to be included in Figure 3 for DR, AMD and glaucoma - outer retinal barrier/blood retinal barrier - transporters - vertical and horizontal pathways (glutamate and GABA transporters and receptors) – excitotoxicity - water and retinal volume regulation – IOP - photoreceptor integrity, dynamics and Vitamin A conversion - trophic support - neurotransmitter cycle – for more information please see doi:10.3390/antiox11040617, Carpi-Santos et al., 2023

11.  Resveratrol fist appears in line 199; but it is abbreviated in line RV 699, it should be abbreviated several times before – please check all

Comments on the Quality of English Language

English is ok

Author Response

REPLAY TO THE REVIEWERS

We thank the reviewers for their valuable comments and suggestions aimed to improve the quality of the manuscript. We modified the manuscript according to the comments of the reviewer and we detail below our modifications, according to their suggestion) and replay to their comments. In the revised version of the manuscript the modifications are underlined in yellow.

Reviewer 1.

We thank this reviewer for the accurate review of our manuscript and the valuable comments. We modified the manuscript according to these comments.

  1. Reviewer: In Abstract Line 10 authors mention 1 billion affected by visual impairment as well as blindness; In line 41, 600 million, while in line 43 - 543 million - please assure an accurate number of patients worldwide affected with the correct reference.

Answer: We modified the abstract in line 10, writing about 600 million. We did not modify line 41, nor line 43 because in line 43 we specified that 43 million show partial or total blindness and about 500 million have poor socio-economic conditions. All the other individuals up to 600 million do not show partial/total blindness, neither a poor socio-economic condition. Thus, visual impairment and medium-high socio-economic conditions.

  1. Reviewer: Lines 62-71 - Authors must develop a better/deeper paragraph related to the highly ordered architecture of the retina containing photoreceptors, which are targeted by several diseases, interneurons, projection neurons (Retinal Ganglion Cell, a target as well, as the optic nerve), and three types of glial cells. Please, describe if the MeDi can ameliorate the structure/function of cell bodies of retinal cells organized in three nuclear layers, cell processes and synapses (Glutamate transporters that avoid excitotoxicity). As the manuscript deals with the retina, and hundreds of millions are affected by visual problems, a schematic figure could illustrate which cells are targeted in different diseases)

Answer: we described the structure of the retina as well as the cell-types/retinal structures altered in age-related macular degeneration, diabetic retinopathy, and glaucoma (lines 79-98 and 100-111). We added a figure describing the cell-types damaged in diabetic retinopathy and the molecular pathways involved (Figure 5). The beneficial effects of the MeDi in the 3 eye diseases has been already described in the corresponding paragraphs. In addition, we focused on the role of Nrf2 pathway and the effect of the MeDi on this pathway. An illustration of the MeDi components involved in the modulation of this pathway is already present (Figure 4).

  1. Reviewer: Lines 73-100 – Mediterranean diet is defined by: whole grains, olive oil, fruits, vegetables, seafood, beans and nuts, which are included in the MeDi, have been linked to the prevention of chronic age-related diseases (AREDs). Are these all the constituents of MeDi? Should a moderate glass of wine be included? Please describe what is in common and what differs from all types of MeDi (Lebanese, Italian and others) in terms of constituents and then jump to the pyramid line 90

Answer: we answered to this point underlying the fact that the Lebanese MeDi shows the major differences compared to the MeDi components of the other countries because red meat, fish and alcohol are not included (lines 132-140). We did not detail further all the differences because they are present in the paragraph dedicated to the description of the various MeDi scoring.

  1. Reviewer: Lines 111-181 – Authors show several important data in a not clear manner…” These differences in the scoring result in an increased difficulty in comparing the data” (I agree). Please try to illustrate information (that are essential to the topic chosen) so the reader finds easier to compare – please use table; differences in terms of quality and healthy benefits of different MeDi (Italian, Lebanese, and others)?  

Answer: we did not include tables describing the components and the “tertiles” of the different MeDi scoring because the detailed analysis of the different MeDi scoring may deserve a dedicated manuscript and is not the central goal of our review. The addition of 7 tables will overload our manuscript, moving the interest far from the major goal of our review that is the effect of the MeDi on the prevention/progression of diabetic retinopathy, age-related macular degeneration and glaucoma. We already summarized the different MeDi scoring system in Figure 2 .

  1. Reviewer: Lines 167-171 reads...In adults, 1 point is added for alcohol intake equivalent to 1 (women) and 2 (men) glasses of wine or beer. Sofi and colleagues developed the MeDi-Lite score in 2017 that analyzed 9 food groups and compared the results with the MDS score, demonstrating that the higher range score provided an increased sensibility and specificity compared to MDS scoring [21]. How can one evaluate the level of resveratrol, alcohol content and the type of grape for the wine added in the scale? For example, Pinot-noir, Malbec, or Carmenere, any difference? Vitis vinifera, labrusca and muscadine types of grapes contain high concentrations of resveratrol (50-100ug/g), but how one can relate to the type of wine chosen?

Answer:  we did not answer to this question because the analysis of the role of resveratrol contained in different wines will need a specific clinical study and a specific questionnaire related to the kind of red wine each individual drink. Moreover, to analyze specifically the effect of resveratrol it is necessary to analyze the effect of specific consumption of resveratrol in a clinical study. Thus, in order to answer to this point it is necessary to carry out a specific clinical study and produce a research article.

  1. Reviewer: Lines 258-270 Linoleic acid (LA) and alpha-linoleic acid (LNA) are essential polyunsaturated fatty acid (PUFA) obtained from several food categories listed in this manuscript (sunflower oil, grape seed oil, safflower oil, walnuts, salmon, chia seeds, sardine, and others 

Answer: we added the list of foods containing LA (lines 314-316).

  1. Reviewer: Several studies demonstrate that the ratio between omega-3 and omega-6 fatty acids is important for the prevention of cardiovascular diseases. Important to mention that LA and LNA are essential lipids obtained in diet through intake of vegetable and animal derived oils…. PLEASE COMMENT ON THE BIOCHEMICAL ROUTES AFTER CONSUMPTION, n-3 and n-6 fatty acids directed to separate pathways for β-oxidation, storage, or elongation. Both LA and LNA undergo several steps of desaturation (Δ6 and Δ5 desaturases) and elongation, generating a great number of metabolites, including ARA, from LA, and EPA and docosahexaenoic (DHA) acids, from LNA. ARA generates the commonly known ECs anandamide (AEA) and 2-AG, while eicosapentaenoyl ethanolamine (EPEA) and docosahexaenoyl ethanolamine (DHEA) are produced from EPA and DHA, respectively, recently recognized as weaker ECs. 

Please mention that DHEA is also known as synaptamide -a trophic factor that improves cognitive parameters in the nervous system. It is a metabolite generated on the omega-3 arm

Answer: we included this part in the revised version of the manuscript adding the appropriate reference (lines 322-330).

  1. Reviewer: Lines 271-275 Dietary intake of omega-3 fatty acids by consumption of fatty fish (IS THIS CORRECT?) or fish oil exerts a beneficial effect by decreasing the risk of cardiovascular diseases, inflammatory bowel disease, cancer, rheumatoid arthritis, psychiatric and neurodegenerative diseases [47]. Omega-6 PUFA is present in extra virgin olive oil. PUFA are contained also in nuts, at high level in pistachio. There are several types of oils that were studied in terms of omega-3/omega-6 contents that could be cited. Please see Iorsavova et al., Int J. Mol. Sci 2015

Answer: we added the vegetal oils containing PUFA in the revised version of the manuscript and the reference of Orsavova et al. (lines 337-339)

  1. Reviewer: Line 294 Saffron is a medicinal plant that has strong anti-inflammatory and antioxidant action, having several health benefits, such as relieving PMS symptoms, helping to control diabetes, promoting weight loss and preventing cardiovascular diseases, for example. Several of the nutraceuticals listed in the manuscript (resveratrol and others found in (1) turmeric root; (2) grapes; (3) green tea; (4) Maytenus senegalensis (Lam.) fruit and roots; (5) black peppers; and (6) blueberries modulate cannabinoid CB1 and/or CB2 receptors, interacting with the endocannabinoid system, enzymes or receptors. Therefore, mechanisms can be explained Curcumin that is present in Saffron and is listed as well.

Answer: we added a brief description of the effects of saffron (lines 359-363) . However, the detailed description of the nutrients and medicinal plants present in the MeDi will deserve a specific review article dedicated to this subject..

  1. Reviewer: Figure 3 - retinal diseases might show complete different set of mechanisms - glaucoma, retinitis pigmentosa, age macular disease, diabethic retinopathy...Authors should specify mechanisms in how the elements/constituents/active principles of MeDi might improve different functions in the retina. I could list several that are important to be included in Figure 3 for DR, AMD and glaucoma - outer retinal barrier/blood retinal barrier - transporters - vertical and horizontal pathways (glutamate and GABA transporters and receptors) – excitotoxicity - water and retinal volume regulation – IOP - photoreceptor integrity, dynamics and Vitamin A conversion - trophic support - neurotransmitter cycle – for more information please see doi:10.3390/antiox11040617, Carpi-Santos et al., 2023

Answer: Figure 3 illustrates the molecular mechanisms that we describe in our manuscript and that are implicated in the three retinal diseases. This figure is not dedicated to illustrating the effect of the MeDi on these diseases. In Figure 4 we illustrate the effect of the Nfr2 pathway in maintain a healthy retina because Nrf2 pathway is considered the major pathway modulated in a positive manner by the MeDi. Indeed, we indicated the role of phytoestrogens in modulating this pathway. We think that it will not be appropriate to add the figure proposed by the reviewer because we did not focus on GABA transporters etc. However, we added figure 5 that describe the alterations present in diabetic retinopathy as example of cellular and molecular alterations in the retina. Following his/her comments, we tried to illustrate additional key mechanisms in the response to hyperglycemia related to retinal diseases. However, it is challenging to explore all the individual biological mechanisms and incorporate them comprehensively into Figure 3. To address this, we have added a new figure highlighting key aspects and integrated the main text of the review with the reference provided, which discusses the essential role of Müller cells.

“Müller cells are particularly noteworthy in the context of retinal diseases, as they play a crucial role in maintaining retinal homeostasis. They regulate various functions, including metabolic support, antioxidant activity, extracellular composition, synaptic activity, neurotransmission, and the structural organization of the blood-retina barrier

  1. Reviewer: Resveratrol fist appears in line 199; but it is abbreviated in line RV 699, it should be abbreviated several times before – please check all

Answer: we corrected and added the abbreviation as suggested by the reviewer.

Reviewer 2 Report

Comments and Suggestions for Authors

The authors of the manuscript ID: 3181706 under the title "Effect of the Mediterranean Diet (MeDi) in the progression of retinal disease, a narrative review" presented interesting information concerning the Mediterranean diet in patients with diabetic retinopathy, macular degeneration and glaucoma

This is a well-prepared review with appropriately cited literature and presented mechanisms. Four figures presenting the effect of the diet on cellular and molecular pathways, including inflammation, immune response, neurodegeneration and consequently leading to retinal diseases, i.e. retinopathy, macular degeneration and glaucoma.

However, the authors did not present the research methodology. Which publications were included and which were excluded in the prepared manuscript.

Inclusion and exclusion criteria are not clearly defined. Why are reviews included? In my opinion, not defining the type of studies can provide important biases in the interpretation of the evidence. I encourage the authors to correct it.

Author Response

REPLAY TO THE REVIEWERS

We thank the reviewers for their valuable comments and suggestions aimed to improve the quality of the manuscript. We modified the manuscript according to the comments of the reviewer and we detail below our modifications, according to their suggestion) and replay to their comments. In the revised version of the manuscript the modifications are underlined in yellow.

Reviewer 2.

Reviewer: The authors of the manuscript ID: 3181706 under the title "Effect of the Mediterranean Diet (MeDi) in the progression of retinal disease, a narrative review" presented interesting information concerning the Mediterranean diet in patients with diabetic retinopathy, macular degeneration and glaucoma

This is a well-prepared review with appropriately cited literature and presented mechanisms. Four figures presenting the effect of the diet on cellular and molecular pathways, including inflammation, immune response, neurodegeneration and consequently leading to retinal diseases, i.e. retinopathy, macular degeneration and glaucoma.

However, the authors did not present the research methodology. Which publications were included and which were excluded in the prepared manuscript.

Inclusion and exclusion criteria are not clearly defined. Why are reviews included? In my opinion, not defining the type of studies can provide important biases in the interpretation of the evidence. I encourage the authors to correct it.

Answer: We thank the reviewer for the valuable comments and suggestions. We added the methods used to collect the data summarized in our manuscript and the number of manuscripts analyzed (lines 61-72). We added this part in the introduction section because our manuscript is a narrative review and it is not a metadata, nor a systematic review. For this reason, we did not include the inclusion and exclusion criteria, which are necessary when writing a metadata analysis or a systematic review.

Round 2

Reviewer 1 Report

Comments and Suggestions for Authors

Authors replied most of the issues raised. It is acceptable